# Metabolic Syndrome and Air Pollution: A Narrative Review of Their Cardiopulmonary Effects

**DOI:** 10.3390/toxics7010006

**Published:** 2019-01-30

**Authors:** Emily A. Clementi, Angela Talusan, Sandhya Vaidyanathan, Arul Veerappan, Mena Mikhail, Dean Ostrofsky, George Crowley, James S. Kim, Sophia Kwon, Anna Nolan

**Affiliations:** 1Department of Medicine, Division of Pulmonary, Critical Care and Sleep Medicine, New York University School of Medicine, New York, NY 10016, USA; Emily.Clementi@nyumc.org (E.A.C.); Angela.Talusan@nyumc.org (A.T.); Sandhya.Vaidyanathan@nyumc.org (S.V.); Arul.Veerappan@nyumc.org (A.V.); Mena.Mikhail@nyumc.org (M.M.); Dean.Ostrofsky@nyumc.org (D.O.); George.Crowley@nyumc.org (G.C.); Jamessungkim320@gmail.com (J.S.K.); Sophia.Kwon@nyumc.org (S.K.); 2Department of Environmental Medicine, New York University School of Medicine, New York, NY 10016, USA; 3Bureau of Health Services and Office of Medical Affairs, Fire Department of New York, Brooklyn, NY 11201, USA

**Keywords:** particulate matter, systemic inflammation, metabolic syndrome, chronic obstructive pulmonary disease, cardiovascular disease, blood pressure, World Trade Center

## Abstract

Particulate matter (PM) exposure and metabolic syndrome (MetSyn) are both significant global health burdens. PM exposure has been implicated in the pathogenesis of MetSyn and cardiopulmonary diseases. Individuals with pre-existing MetSyn may be more susceptible to the detrimental effects of PM exposure. Our aim was to provide a narrative review of MetSyn/PM-induced systemic inflammation in cardiopulmonary disease, with a focus on prior studies of the World Trade Center (WTC)-exposed Fire Department of New York (FDNY). We included studies (1) published within the last 16-years; (2) described the epidemiology of MetSyn, obstructive airway disease (OAD), and vascular disease in PM-exposed individuals; (3) detailed the known mechanisms of PM-induced inflammation, MetSyn and cardiopulmonary disease; and (4) focused on the effects of PM exposure in WTC-exposed FDNY firefighters. Several investigations support that inhalation of PM elicits pulmonary and systemic inflammation resulting in MetSyn and cardiopulmonary disease. Furthermore, individuals with these preexisting conditions are more sensitive to PM exposure-related inflammation, which can exacerbate their conditions and increase their risk for hospitalization and chronic disease. Mechanistic research is required to elucidate biologically plausible therapeutic targets of MetSyn- and PM-induced cardiopulmonary disease.

## 1. Introduction

Cardiopulmonary disease caused by ambient particulate matter exposure accounts for 7 million deaths globally each year [1,2,3]. Epidemiologic associations have been documented between increased ambient particulate matter (PM), lung disease, and cardiovascular disease (CVD) [4,5,6,7,8,9,10,11]. The aim of this review article is to provide an up-to-date overview of the epidemiological and biological mechanism of PM-induced systemic inflammation in MetSyn, obstructive lung disease, and CVD. This review also discusses the contribution of PM exposure and MetSyn to cardiopulmonary disease. A cohort of firefighters that was exposed to World Trade Center Particulate Matter (WTC-PM) has been identified as having metabolically active biomarkers associated with the development of WTC-lung injury (WTC-LI) [12,13,14,15,16,17,18]. We also specifically discuss the relationship between PM exposure and MetSyn in the WTC-exposed FDNY firefighters.

## 2. Review/Search Strategy

PubMed databases were searched in July and August of 2018. The search was also limited to articles that were published within the last 16 years, from January 2002 to August 2018. Cohort studies, case control studies, narrative reviews, meta-analyses, and mechanistic and statistical summaries were retrieved. Titles, abstracts, and full texts were screened based on relevance to this review. Keywords searched included: “obstructive airway disease”, “asthma”, “air pollution”, “particulate matter”, “occupational exposure”, “World Trade Center”, and “chronic obstructive pulmonary disease.” In addition, the references of many of the articles identified by the above search strategy were reviewed.

### Inclusion/Exclusion Criteria and Limitations

Studies were included in our narrative review if they: (1) described the coexistence of MetSyn, cardiopulmonary disease and PM exposure, (2) detailed the mechanisms of PM-induced inflammation within these diseases, or (3) focused on the effects of PM-exposure in WTC-exposed FDNY firefighters. We included (4) observational, retrospective, systematic reviews or clinical studies that focused on: (5) providing the epidemiology and etiology of PM and associated MetSyn and cardiopulmonary disease, or (6) the use of biomarkers to evaluate environmentally associated MetSyn and cardiopulmonary disease.

Studies were excluded if they: (1) were not included in PubMed, (2) were published earlier than 2002, or (3) were not written in the English language. Studies included in this review were available in their entirety online and were referenced using Endnote X8 (Thomson Reuters, Philadelphia, PA, USA, 2016).

Limitations of this study design include: (1) use of only the PubMed database, (2) the fact that it is not a systematic review, and therefore (3) performing a full meta-analysis of the obtained data is limited.

## 3. Epidemiological Studies

### 3.1. Epidemiology of MetSyn and PM Exposure

MetSyn is defined as having at least three of five risk factors associated with development of cardiovascular disease, diabetes, and stroke: abdominal obesity, insulin resistance, hypertriglyceridemia, low HDL, and hypertension, Figure 1 [19].

PM exposure, having been linked to developing elevated blood pressure (BP), is a risk factor for developing MetSyn characteristics, Figure 1a. PM exposure is often focused on the respirable portion of ambient air (2.5 and 10 micrometers in size). In a recent longitudinal study, adults that resided in locations with high ambient PM_2.5_ concentrations experienced significant elevations in diastolic BP. Overweight adults living in the same area experienced increases in both systolic and diastolic BP; however, no PM_2.5_-related BP changes were found in locations where ambient PM_2.5_ concentrations remained low; therefore, this study suggests increased PM_2.5_ exposure promotes elevations in BP among healthy and obese individuals, with the latter being more susceptible to the effects of ambient air pollution [20].

Additional human studies have demonstrated that PM_2.5_ causes an increase in BP after only a few days of exposure, and exposure over years can lead to chronic hypertension, Figure 1a,d [21]. PM inhalation also promotes the development of insulin resistance, which has been mechanistically linked to hypertension [22]. Insulin resistance, is considered the primary risk factor for diabetes mellitus [23]. A recent longitudinal study of 1,729,108 participants followed for a median of 8.5 years showed that PM_2.5_ air pollution is significantly associated with an increased risk of diabetes, Table 1 [24].

Furthermore, individuals with preexisting MetSyn are more susceptible to the inflammatory effects of PM exposure [2,32,36]. Chronic exposure to ambient air pollution leads to weight gain secondary to local and systemic inflammation, increasing the risk of developing the etiological components of MetSyn [37,38]. As MetSyn affects more than 30% of adults in the United States, patients with MetSyn represent a large percentage of the population that is especially sensitive to PM [2,32,39,40].

### 3.2. Epidemiology of CVD and PM Exposure

PM exposure has been linked to an increased risk of CVD such as myocardial infarction, ischemic heart disease, stroke, heart failure, arrhythmias, and venous thromboembolism (Figure 1e [41]). Short-term PM exposure was associated with an increased number of hospital admissions for CVD, and both fatal and non-fatal cardiovascular events [34]. Long-term exposure is associated with an even greater increase of cardiovascular disease and mortality, including postmenopausal women from U.S. metropolitan areas (Table 1 [35,41]).

### 3.3. Epidemiology of Chronic Obstructive Pulmonary Disease (COPD) and PM Exposure

PM exposure can elicit the development or exacerbation of COPD (Figure 1e). Black carbon, an indicator of traffic-related fine particulate air pollution, was associated with an increased risk for COPD hospitalization in a population-based study of 467,994 subjects [27]. An increase of 10 µg/m^3^ in PM_2.5_ nearly doubled the hospital admissions for COPD exacerbations from 1999 to 2002 in a study of Medicare billing claims from 11.5 million enrollees [10]. Similar to individuals with preexisting MetSyn, COPD patients are more susceptible to the harmful effects of PM exposure and often experience acute exacerbations due to bacterial and viral infections contracted in the wake of PM exposure [42,43]. Additionally, long-term PM exposure has been implicated as a potential indicator of increased respiratory mortality among COPD patients [43]. In a 2014 cross-sectional study, it was determined that there was a 2.53% increase in COPD deaths per 10 µg/m^3^ increase of PM_2.5_ over a six-day period, Table 1 [29].

## 4. Biological Mechanisms Underlying PM-Induced Metabolic and Cardiopulmonary Diseases

### 4.1. Mechanisms of PM Associated MetSyn

Air pollution has been implicated in the pathogenesis of MetSyn by causing systemic inflammation associated with metabolic disorders [25]. This chronic inflammation is characterized by cytokine production, and activation of a network of inflammatory signaling pathways. Adipose tissue is involved in the inflammatory response and mediators [44]. Tumor Necrosis Factor-alpha (TNF-α) is a pro-inflammatory cytokine that is overexpressed in the adipose tissue of obese mice and humans after PM exposure [45,46]. Cytokines such as TNF-α and Interleukin-6 (IL-6), lipids, reactive oxygen species (ROS), or endoplasmic reticulum (ER) stress activate various signal transduction cascades by inducing the activity of cellular kinases, namely c-Jun N-terminal Kinase (JNK), I-kappa B kinase (IKK), and Protein Kinase C (PKC) (Figure 1b [36,44]). These kinase phosphorylate serine residues of insulin receptor substrate-1 and -2 in order to block insulin action. JNK and IKK also promote further inflammatory gene expression by activating the two principal inflammatory pathways: activator protein 1 (AP-1) and nuclear factor kappa-B (NF-κB), respectively [44]. Adipocyte hypertrophy in response to fat consumption and accumulation can induce cellular rupture, attracting macrophages to reinforce the inflammatory response. In this context, hypertrophied adipocytes rupture frequently, leading to the deposition of fat in organs other than adipose tissue, Figure 1b [36].

In animal models, air pollution has been linked to hypertension, alterations in blood lipids, insulin resistance, and obesity, all of which contribute to the low-grade systemic inflammation of MetSyn (Figure 1d). Pregnant rats exposed to unfiltered Beijing air for 19 days (starting on gestational day 1), starting on their first day of gestation, were heavier at the end of their pregnancy compared to those who were exposed to filtered air. Additionally, 8-week-old pups who were prenatally and postnatally exposed to unfiltered air were significantly heavier than those who were exposed to filtered air. The PM-exposed pups also demonstrated significantly lower levels of Glucagon-like Peptide 1 (GLP-1), an incretin hormone that enhances insulin secretion and has anti-inflammatory properties within adipose tissue. Both the previously pregnant rats and the 8-week-old pups displayed perivascular and peribronchial inflammation in the lungs. Particulate matter caused PM-exposed rats to experience weight gain secondary to systemic inflammation, increased insulin resistance and lung inflammation, which are etiological components of MetSyn. This study suggests that chronic exposure to particulate matter increases the risk of developing MetSyn [25]. In a murine model, exposure to PM_10_ showed elevated neutrophil concentrations and upregulated TNF-α and IL-6 levels, all of which indicate an inflammatory response. Additionally, mice exposed to PM experienced an upregulation of genes related to inflammation, cholesterol and lipids, Figure 1a [1]. In another murine model, mice that were fed high-fat chow for 10 weeks and exposed to 72.7 μg/m^3^ of PM_2.5_ for 6 h/day, 5 days/week over 24 weeks experienced exaggerated insulin resistance, systemic inflammation, and visceral adiposity demonstrated by elevated TNF-α, IL-6 and PKC expression, Figure 1b [26,37].

In humans, ambient PM has also been found to induce DNA hypomethylation, which is associated with increases in BP, Figure 1a. Hypomethylation gives rise to vascular smooth muscle proliferation and lipid deposition due to mutations, causing the formation of fibrocellular lesions and subsequent increases in BP [47]. Autonomic imbalance has also been suggested as a possible mechanism through which PM increases diastolic BP, a component of MetSyn, Figure 1a [19,21,48]. Exposure to PM_2.5_ lowers repetitive element *Arthrobacter luteus* (*Alu*) methylation, while exposure to PM_2.5–10_ lowers toll-like receptor 4 (*TLR4*) methylation. Both *Alu* and *TLR4* hypomethylation are associated with increased diastolic BP, while only *TLR4* hypomethylation is associated with increased systolic BP, Figure 1a [47].

### 4.2. PM Exposure and COPD

Inflammation and tissue remodeling are key features of airflow obstruction in asthma and obstructive airways disease (OAD), as discussed in two reviews [49,50]. PM exposure leads to the pathogenesis of COPD by inducing pulmonary and systemic inflammation (Figure 1e). Chronic exposure to air pollution can prevent clearance of PM from the lung, resulting in particle accumulation in lung tissues. The accumulation of PM in the respiratory tract induces the production of pro-inflammatory mediators, namely TNF-α and IL-6, by alveolar macrophages and lung epithelial cells (Figure 1b [43]).

Inflammatory cytokines that increase in sputum and bronchoalveolar lavage fluid are also elevated in COPD patients, suggesting inflammatory mediators elicit both a local inflammatory response in the lung tissues and secondary systemic inflammatory response [51]. The inflammatory response is characterized by tissue proliferation in the small airways and tissue destruction in the lung parenchyma, causing subsequent airway obstruction, leading to the development of COPD [43].

### 4.3. PM Exposure and CVD

Recent studies have associated systemic vascular dysfunction with lung disease (the vascular hypothesis) and prospective studies have demonstrated an association between impaired lung function and central arterial stiffness even before the development of CVD [52,53,54,55,56]. Hallmarks of PM exposure include vascular endothelial dysfunction, systemic inflammation, and subsequent end-organ damage [13,57,58,59,60,61,62,63].

In a murine model, the effects of long-term PM exposure on atherosclerosis potentiation, vasomotor tone alteration, and vascular inflammation were evaluated. This study found that PM-exposed and high-fat chow fed mice exhibited significant plaque burden, compared to PM-exposed mice with a normal chow diet [31]. Another murine study demonstrated that acute carbon black exposure led to impaired cardiac function in senescent mice through cardiac changes such as diminished myocardial contractibility, elevated right atrial and pulmonary vascular pressures, and increased pulmonary vascular resistance [30]. 

In humans, three PM-related pathways have been linked to adverse cardiovascular health effects: (1) the generation of systemic inflammation through the release of circulating pro-inflammatory and pro-oxidative mediators from PM-stimulated lung cells (Figure 1b), (2) alterations in cardiac autonomic function induced by PM interactions with lung receptors (Figure 1a), and (3) the translocation of PM into the bloodstream [33,41,64]. PM inhalation triggers local and systemic inflammation through these biological mechanisms depending on the size of the PM. Both coarse (PM_10–2.5_) and fine (PM_2.5_) PM can trigger the release of inflammatory mediators that spread to general circulation, where they elicit systemic inflammation [41]. Generation of ROS is also involved in the PM-induced pro-inflammatory pathway as demonstrated by elevated ROS in rat lung and heart after PM exposure (Figure 1b [65]). ROS have been linked to atherosclerosis, vascular dysfunction, cardiac arrhythmias, and myocardial injury [66]. Coarse and fine particles also cause impairment of the autonomic control of the heart, by enhancing sympathetic tone and decreasing heart rate variability (Figure 1a [41,67]). Both decreased heart rate variability, an indicator of poor cardiovascular prognosis, and elevated sympathetic tone predispose individuals to arrhythmia-associated cardiac death [66,67]. Ultrafine particles (PM < 0.1 µm) can translocate into the bloodstream, where they promote events by enhancing platelet aggregation and endothelial cell activation [41]. Furthermore, not only does PM exposure put individuals at risk for the development of CVD, but that MetSyn phenotypes also influence these pathways’ differential response to PM exposure.

## 5. MetSyn as a Risk Factor for COPD and CVD

PM-induced systemic inflammation and co-existing MetSyn have been implicated in the development and progression of cardiopulmonary diseases, Figure 1e [33,37,38,60,68,69]. Individuals with MetSyn are predisposed to systemic inflammation, a key feature of COPD [28], and nearly half of COPD patients have coexisting MetSyn [68,70,71]. A cross-sectional study demonstrated that systemic inflammatory markers were elevated in COPD patients with preexisting MetSyn, compared to those without MetSyn [28]. This study suggests that systemic inflammation is more severe in patients with coexisting COPD and MetSyn than in healthy individuals [28,72]. Additionally, systemic inflammation contributes to the development of cardiovascular disease, reaffirming that the concurrence of MetSyn and COPD increases the risk of cardiovascular morbidity and mortality [28]. 

Individuals with MetSyn are especially susceptible to the cardiovascular effects of air pollution [73]. Exposed individuals with MetSyn experience increased oxidative stress, which is further elevated by aromatic hydrogen and metal nanoparticle components of air pollution. Consequently, an oxidative stress cascade is activated, leading to CVD (Figure 1b,c [38]). In a case-crossover study, MetSyn individuals with no preexisting CVD who were exposed to ambient ultrafine particles experienced PM-induced cardiovascular effects, demonstrated by changes in heart rate variability and cardiac repolarization [32]. Similarly, a population-based study reported that, after PM_2.5_ exposure, those with MetSyn exhibited substantial decreases in heart rate variability relative to those without MetSyn; therefore, PM exposure increases cardiovascular risk among MetSyn patients with or without cardiovascular disease [33].

## 6. Cardiopulmonary Effects of WTC-PM-Exposure

Our group has developed leadership in pathophysiologic investigation of WTC-associated disease [16,17,18]. FDNY rescue/recovery workers exposed to WTC-PM developed respiratory symptoms and were diagnosed with chronic pulmonary diseases, including OAD and airway hyperreactivity [15,39,74,75,76,77,78,79,80,81,82,83,84,85]. Induced sputum drawn from WTC-exposed FDNY firefighters 10-months post exposure showed elevated levels of PM and evidence of continuing inflammation due to abnormal accumulation of pro-inflammatory cells [84,86]. The WTC-exposed FDNY cohort experienced significant decreases in Forced Expiratory Volume in 1 second (FEV_1_) [51,87].

In addition to the pulmonary effects of WTC exposure, there was an increase in the risk for CVD-related hospitalizations post WTC exposure [88,89]. CVD symptoms such as chest pain were found in 8% of WTC-exposed workers and volunteers between 2002 and 2004 [90]. Furthermore, pulmonary arteriopathy was present in 58% of lung biopsies from a small group of WTC-exposed individuals [91].

Biomarkers of MetSyn, traditionally seen as risk factors for CVD, predict WTC-associated OAD [13,14,92]. Specifically, BMI-adjusted triglycerides, HDL, heart rate, and leptin were significantly elevated, indicating that metabolic risk factors held key roles in the inflammatory cascade from PM exposure [13]. Also, our study of computed tomography scans of WTC firefighters showed that elevated Pulmonary-Artery-to-Aorta diameters ratio (PA/A) is correlated with future development of FEV_1_ [12].

Our investigation of the metabolome of WTC-associated OAD has identified prominent pathways involving lipids in the same exposed firefighters [12]. Pathological imbalances in lipid metabolism are well-defined initiators of systemic inflammation, triglyceridemia, CVD, and OAD. Aspects of our cohort’s lipid metabolome that have been correlated with OAD include arachidonic acid, lysophosphatidic acid (LPA), lysolipids, phospholipids, polyunsaturated fatty acids, and phosphatidylcholines (1-palmitoyl-2-arachidonoyl-sn-glycero-3-phosphocholine and 1-stearoyl-2-arachidonoyl-sn-glycero-phosphocholine) [12]. Studies have connected these intermediates to a cascade initiated by ROS production, and culminating in triglyceride production and systemic inflammation, Figure 1c [12]. The association of ROS production and lipid imbalance yields a PM-initiated pathway of catabolism, resulting in lipid-mediated inflammation, CVD, and COPD [12].

Our recent work has focused on the receptor for an advanced glycation end products (RAGE)/LPA axis [12]. Our collaborators have identified a ligand-receptor interaction between LPA and the advanced glycation end-product receptor (RAGE), a cytoplasmic IgG receptor localized to alveolar macrophages, alveolar endothelium, and smooth muscle within lung tissue [12]. Specifically, we have shown that elevated soluble RAGE and LPA are associated with WTC-LI in firefighters exposed to WTC-PM and mice are deficient in RAGE are protected from the adverse pulmonary effects [12].

## 7. Conclusions and Future Investigations

Overall, we found that exposure to particulate matter elicits pulmonary and systemic inflammation. Systemic inflammation leads to the development of MetSyn and cardiopulmonary disease, such as COPD and CVD. Individuals with these preexisting conditions are more susceptible to the inflammatory effects of PM exposure, which can further exacerbate their conditions. Additionally, MetSyn predisposes individuals to PM-induced pathogenesis of COPD and CVD; therefore, further research is required to discover and elucidate therapeutic targets of these comorbidities.

## Figures and Tables

**Figure 1 toxics-07-00006-f001:**
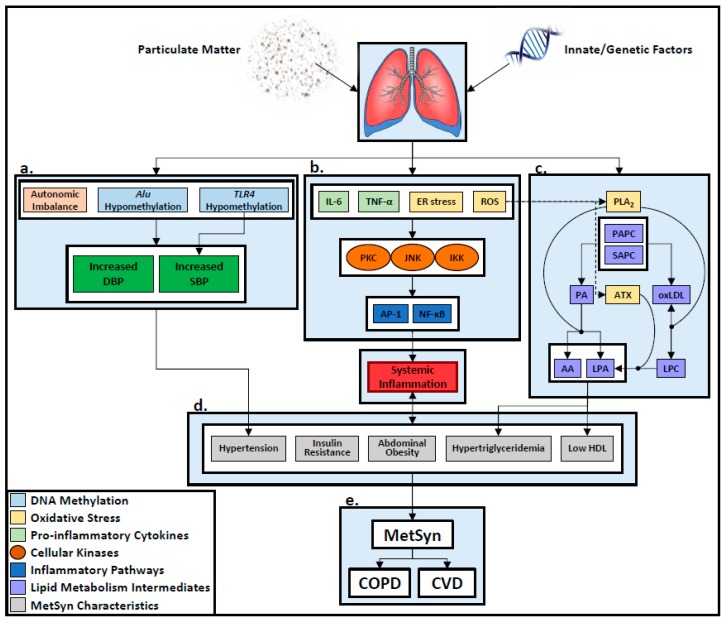
Schematic of biological mechanisms underlying PM-induced MetSyn, COPD, and CVD. (**a**) PM- associated Vascular Effects: PM exposure leads to the hypomethylation of TLR4, which may increase systolic blood pressure (SBP) and diastolic blood pressure (DBP), as well as *Alu* hypomethylation and autonomic imbalance, which may elevate DBP. Increased SBP and DBP contribute to the development of hypertension, a key feature of MetSyn. (**b**) PM associated MetSyn Phenotype Development: Inhalation of PM elicits the generation of reactive oxygen species (ROS), ER stress, and elevated cytokine levels, such as TNF-α, and IL-6, which in turn activates signal transduction cascades by inducing the activity of cellular kinases (JNK, PKC, IKK). Kinase activation can directly lead to systemic inflammation or do so indirectly by first activating inflammatory pathways (AP-1, NF-κB). (**c**) PM-associated Lipid Changes (ATX Autotaxin; LPC lysophosphatidylcholine; PLA phospholipase; PAPC 1-palmitoyl-2-arachidonoyl-sn-glycero-3- phosphocholine; SAPC 1-stearoyl-2-arachidonoyl-sn-glycero-phosphocholine; oxLDL oxidized LDL; PA phosphatidic acid). (**d**) Systemic Inflammation contributes to the development of insulin resistance, abdominal obesity, hypertriglyceridemia, and low HDL, all of which are defining characteristics of MetSyn. (**e**) MetSyn then increases affected individuals’ risk of developing COPD and CVD. Lines with no arrowhead (
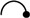
) indicate enzymatic contribution to downstream catabolic reactions.

**Table 1 toxics-07-00006-t001:** Overview of the effects of PM exposure on metabolic and cardiopulmonary diseases.

Disease	Study	Country	Study Population	Significant Findings
METSYN	Animal Studies			
	Brocato [1]	USA	Murine model	PM exposure enhances the expression of genes located in pathways associated with MetSyn.
	Wei [25]	China	Murine model	Chronic exposure to PM increases the risk of MetSyn.
	Sun [26]	USA	Murine model	Long-term PM exposure exacerbates MetSyn.
	Human Studies			
	Huang [20]	China, USA	Longitudinal cohort	High PM_2.5_ exposure promotes BP elevations in healthy and overweight individuals.
	Bowe [24]	USA	Longitudinal cohort	Inhalation of PM_2.5_ is significantly associated with increased risk for developing diabetes mellitus (HR, 1.15; 95% CI, 1.08–1.22).
	Naveed [13]	USA	Longitudinal cohort	MetSyn biomarkers—abnormal triglycerides and HDL (OR, 3.03; 95% CI, 1.39–6.16) and elevated heart rate (OR, 2.20; 95% CI, 1.14–4.24) and leptin (OR, 3.00; 95% CI, 1.35–6.66)—are risk factors of lung function impairment after WTC PM exposure.
COPD	Human Studies			
	Gan [27]	Canada	Population-based cohort	Exposure to particulates in traffic-related air pollution was associated with a 6% increase in the risk of COPD hospitalization (95% CI, 2–10%).
	Dominici [10]	USA	Population-based cohort	Increased PM exposure doubled hospital admissions for COPD exacerbations.
	Vujic [28]	Serbia	Cross-sectional	Systemic inflammatory markers are higher in COPD patients with MetSyn than in those without MetSyn. Individuals with MetSyn have a higher leukocyte count (OR, 1.321; 95% CI, 1.007–1.628) and C-reactive protein level (OR, 1.184; 95% CI, 1.020–1.376) compared to those without MetSyn.
	Samoli [29]	Europe	Cross-sectional	PM_2.5_ is positively associated with mortality due to diabetes (1.23%; 95% CI, 1.63–4.17%), cardiac causes (1.33%; 95% CI, 0.27–2.40%), COPD (2.53%; 95% CI, 0.01–5.14%), and to a lesser degree to cerebrovascular causes (1.37%; 95% CI, 1.94–4.78%).
CVD	Animal Studies			
	Tankersley [30]	USA	Murine model	Carbon black exposure led to impaired cardiac function in senescent mice
	Sun [31]	USA	Murine model	Long-term PM exposure altered vasomotor tone, induced vascular inflammation, and potentiated atherosclerosis.
	Human Studies			
	Devlin [32]	USA	Case-crossover	MetSyn patients with no overt CVD experienced PM-induced cardiovascular changes.
	Park [33]	USA	Longitudinal cohort	As a result of PM exposure, individuals with MetSyn had significantly larger decreases in heart rate variability measures than those without MetSyn. Patients with MetSyn experienced a 2.1% decrease in the root mean square of successive differences (95% CI, −4.2–0.0) and a 1.8% decrease in the standard deviation of normal-to-normal intervals (95% CI, −3.7–0.1).
	Chang [34]	Taiwan	Case-crossover	Short-term PM exposure increases hospital admissions for CVD. On cool days, PM_2.5_ exposure was associated with a 47% (95% CI, 39–56%), 48% (95% CI, 40–56%), 47% (95% CI, 34–61%), and 51% (95% CI, 34–70%) increase in ischemic heart disease, stroke, congestive heart failure, and arrhythmias hospital admissions, respectively.
	Miller [35]	USA	Prospective cohort	Long-term PM exposure was related to cardiovascular disease and mortality. Each increase of 10 microgram per cubic meter of PM_2.5_ was associated with a 24% increase in the risk of cardiovascular event (HR, 1.24; 95% CI, 1.09–1.44) and a 76% increase in the risk of death from CVD (HR, 1.76; 95% CI, 1.25–2.47).

Abbreviations: MetSyn Metabolic Syndrome; CI Confidence Interval; COPD Chronic Obstructive Pulmonary Disease; PM Particulate Matter; CVD Cardiovascular disease; HDL High Density Protein; HR Hazards Ratio; OR Odds Ratio; USA United States of America. PM_2.5_ Particulate Matter <2.5 µm in Aerodynamic Diameter.

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
