# Peer review of "Metabolic Syndrome and Air Pollution: A Narrative Review of Their Cardiopulmonary Effects"

_toxics, 2019, doi:10.3390/toxics7010006_

Round 1
Reviewer 1 Report
Review/Search Strategy: the statements used in Pubmed should be clearly reported. To do an example: probably the authors searched “World trade center” and not World trade center (single words). Furthermore, authors should be aware that some articles may be not included in Pubmed. It should be indicated as a limitation of the study.
Table 1: I would include the observed effects (i.e. OR, HR, etc) with confidence interval, if available.
General organization: I would distinguish the effects on animals and humans in each paragraph, to underline what is observed on animals and the parallel evidences on humans. Authors may divide each paragraph in two (I suggest the same with table 1).
Final suggestion (not mandatory to accept the manuscript): are there some available data that could be analyzed with meta-analysis?
Author Response
Please See Attached Word Document

Reviewer 2 Report
PubMed databases were searched in July and August of 2018 with search was limited to articles that were published within the last 16 years. studies included Cohort studies, case control studies, narrative reviews, meta‐analyses, and mechanistic and statistical summaries were retrieved. The literature review is extensive.
The mechanisms are discussed and the article is therefore up to date, and a useful reference article.
The multiple abbreviations makes the readability heavy going at times. The WTC data sets were new to me and of interest. That short term exposure could have long term complications, applied to other than respiratory conditions is of interest.
Author Response
Please see attached word document for response to reviewer 2
